# Removal of the Polyglutamine Repeat of Ataxin-3 by Redirecting pre-mRNA Processing

**DOI:** 10.3390/ijms20215434

**Published:** 2019-10-31

**Authors:** Craig S. McIntosh, May Thandar Aung-Htut, Sue Fletcher, Steve D. Wilton

**Affiliations:** 1Molecular Therapy Laboratory, Centre for Molecular Medicine and Innovative Therapeutics, Murdoch University, Health Research Building, Discovery Way, Murdoch WA 6150, Australia; C.McIntosh@murdoch.edu.au (C.S.M.); M.Aung-Htut@murdoch.edu.au (M.T.A.-H.); s.fletcher@murdoch.edu.au (S.F.); 2Perron Institute for Neurological and Translational Science, Centre for Neuromuscular and Neurological Disorders, The University of Western Australia, Nedlands WA 6009, Australia

**Keywords:** spinocerebellar ataxia type 3, antisense oligonucleotides, exon skipping, ataxin-3, polyglutamine, phosphorodiamidate morpholino oligomer

## Abstract

Spinocerebellar ataxia type 3 (SCA3) is a devastating neurodegenerative disease for which there is currently no cure, nor effective treatment strategy. One of nine polyglutamine disorders known to date, SCA3 is clinically heterogeneous and the main feature is progressive ataxia, which in turn affects speech, balance and gait of the affected individual. SCA3 is caused by an expanded polyglutamine tract in the ataxin-3 protein, resulting in conformational changes that lead to toxic gain of function. The expanded glutamine tract is located at the 5′ end of the penultimate exon (exon 10) of *ATXN3* gene transcript. Other studies reported removal of the expanded glutamine tract using splice switching antisense oligonucleotides. Here, we describe improved efficiency in the removal of the toxic polyglutamine tract of ataxin-3 in vitro using phosphorodiamidate morpholino oligomers, when compared to antisense oligonucleotides composed of 2′-*O*-methyl modified bases on a phosphorothioate backbone. Significant downregulation of both the expanded and non-expanded protein was induced by the morpholino antisense oligomer, with a greater proportion of ataxin-3 protein missing the polyglutamine tract. With growing concerns over toxicity associated with long-term administration of phosphorothioate oligonucleotides, the use of a phosphorodiamidate morpholino oligomer may be preferable for clinical application. These results suggest that morpholino oligomers may provide greater therapeutic benefit for the treatment of spinocerebellar ataxia type 3, without toxic effects.

## 1. Introduction

Spinocerebellar ataxia type 3 (SCA3) is a progressive, typically late-onset autosomal dominant neurodegenerative disease [1]. SCA3 is one of a larger group of diseases, termed, the polyglutamine (polyQ) diseases [2,3]. These diseases all share a common pathogenic mechanism; an expanded CAG repeat in the coding sequence of nine genes, and in the case of SCA3, the CAG expansion is located in the penultimate exon (exon 10) of *ATXN3* (14q32.1) [4]. Healthy individuals have a stable repeat range of 7–44, while SCA3 patients usually have 54 or more repeats. SCA3 is known to have an unstable pre-mutation range of 45–53 repeats, and while these individuals are typically asymptomatic, they have the ability to pass on an expanded allele in what is known as ‘genetic anticipation’. As with other polyQ diseases, the pathogenic severity and age of onset is typically inversely correlated to the size of the expansion: the larger the expansion, the more severe the pathogenesis and the earlier the age of onset [5]. The *ATXN3* encodes for a 361 amino acid (aa), 45 kDa protein (ENST00000558190.6), termed ataxin-3. The ataxin-3 protein is known to act as an isopeptidase and is well documented in cell deubiquitination, as well as proteasomal protein degradation [2,6]. 

The expanded CAG repeat located in exon 10 of *ATXN3* results in the addition of an extended glutamine tract in ataxin-3, directly leading to conformational changes that give the protein a toxic gain of function(s), as well as subjecting the protein to formation of neuronal nuclear inclusions [7]. Although SCA3 is clinically heterogeneous in presentation, the main feature is progressive ataxia, which in turn affects speech, balance and gait of the affected individual [3]. Despite arising from a single variant gene, the pathogenesis of SCA3 has been difficult to characterize, as several toxic pathways and mechanisms have been proposed to play a role in the disease.

Several studies that use antisense oligonucleotides (AOs) to modify the mRNA of *ATXN3* by attempting to remove the CAG containing exon have been conducted [8,9,10]. Until now, van Roon-Mom and colleagues have published two reports detailing the removal of the CAG containing exon in the *ATXN3* transcript [8,9]. These studies show removal of the CAG containing exon, and production of a functional truncated protein using a modified 2′-*O*-methoxy-ethyl nucleotide (2′-MOE) on a phosphorothioate (PS) backbone. Growing concerns regarding safety of PS oligonucleotides, such as hepatotoxicity, liver necrosis and altered regulation of protein and metabolic pathways [11,12,13] and thus, the consequences and safety of long-term exposure to AOs on a PS backbone [11,12,13,14,15,16] may limit applicability of these compounds. Phosphorothioate backbone AOs have resulted in severe injection site reactions, thrombocytopenia and renal toxicity in clinical studies [16,17]. An in vitro study by Flynn et al. (2018), demonstrated severe, sequence-independent backbone-specific effects of 2′-*O*-methyl modified bases on a phosphorothioate backbone (2′-Me PS) AOs, including altered distribution of nuclear proteins, the appearance of abnormal but highly structured nuclear inclusions and aggregates and global disturbance of the transcriptome [18].

An AO chemistry currently in clinical use is the phosphorodiamidate morpholino oligomer (PMO) [19]. This chemistry has a backbone of methylene morpholine rings, with phosphorodiamidate group linkages and an oligomer of this chemistry has been granted accelerated approval by the FDA. *Eteplirsen* (Sarepta Therapeutics, Ma) is designed to excise dystrophin exon 51 during pre-mRNA processing to restore functional protein expression in a subset of boys with Duchenne muscular dystrophy (DMD). The drug restored modest dystrophin expression in patient muscle where previously no, or only traces of dystrophin were evident [20]. The PMO chemistry is reported to have excellent biological stability, and to be safe and well tolerated, with no serious adverse effects reported in the *Eteplirsen* treated children and young men to date [21,22]. 

Here, we describe efficient removal of the CAG containing exon 10 to produce a truncated ataxin-3 protein, lacking the polyglutamine tract, an isoform reported by Toonen et al. (2017) to be functionally active [8]. Our study shows that by using the PMO chemistry, not only is exon 10 skipping enhanced at the RNA level, but also significant downregulation of the protein with higher number of glutamine repeats and an increase in production of the truncated protein is observed, when compared to the use of the 2′-Me PS AO chemistry. With robust splice switching efficiency and an established long-term safety profile, the PMO oligomers described here are presented as lead pre-clinical candidates to treat SCA3 patients.

## 2. Results

### 2.1. ATXN3 Transcript and Strategic Removal of Exons

The predominant full-length *ATXN3* transcript (ENST00000558190.6) includes 11 exons and is approximately 7000 bases in length (Figure 1) and encodes the 361 aa ataxin-3 (Figure 1). The initial focus of this study was to utilise splice switching AOs to remove the polyQ containing exon from the mRNA transcript and thus create an internally truncated protein, missing the toxic polyQ tract. AOs were designed to remove exons 9 and 10 in order to keep the reading frame intact, with the locations of the AO annealing sites illustrated in Figure 1. Removal of the polyQ tract as a therapeutic strategy is plausible, as the main functional domain (Josephin Domain) is located at the N-terminus of the protein, encoded by exons 1–7. Other vital functional domains include the ubiquitin interacting motifs (UIM1–3), as well as the nuclear localisation signal (Figure 1). 

### 2.2. Evaluation of AOs to Induce Exon 9 and 10 Skipping from the ATNX3 Transcript

Initially, several 2′-Me PS AOs were designed to target exon 9 and exon 10 simultaneously for removal from the *ATXN3* transcript (Table 1), in an effort to exclude the polyQ domain and maintain the open reading frame of the transcript and the entire 3′UTR. An unrelated control AO that does not anneal to any transcript was included in all transfections as a negative experimental control. All AOs were transfected into SCA3 patient-derived dermal fibroblasts (74Q; 24Q) at various concentrations (400, 200, 100 nM) for 24 h before RNA extraction and RT-PCR to assess exon skipping. Gel fractionation of the RT-PCR amplicons revealed shorter products arising from exclusion of either exon 9 (Δ97 bp) or exon 10 (Δ119 bp) (Figure 2A), confirmed by Sanger sequencing (Figure 2B). Efficient exon skipping was achieved with AOs targeting exon 9 at concentrations as low as 100 nM.

Excising exon 10 alone removes the polyQ coding motif from the transcript and introduces a novel stop codon early in exon 11, resulting in the loss of 69 aa from the C-terminus of ataxin-3 (ENST00000558190.6). The consequence of excluding this block of amino acids leads to the loss of the UIM3 domain (Figure 2C,D). However, this exact isoform has been reported by Toonen et al. (2017) and from their data, it appeared that the loss of UIM3 does not adversely affect the ubiquitin binding capability of ataxin-3 [8]. The most efficient exon skipping was induced at a transfection concentration of 400 nM, with little cell death evident. At transfection concentrations of 600nM and greater, toxicity associated with 2′-Me PS AO cationic lipoplexes caused almost 100% cell death (data not shown).

### 2.3. PMO Mediated Exon Skipping Reduces Full-Length Ataxin-3 Proteins and Induces a Truncated Ataxin-3 Isoform

Following initial screening of AOs to remove exon 10, this strategy was deemed most appropriate due to the need to only skip one exon to generate a functional ataxin-3 isoform. Equimolar amounts of two AOs targeting *ATXN3* exon 10 were combined into cocktails and evaluated, as we have found that the efficiency of exon removal can be greatly enhanced by such strategies (Table 2) [23,24]. All cocktails reported were co-transfected at equal molar AO (1:1) ratios. The transfection concentration of 400 nM (200 nM of each AO) was selected and as expected, some AO cocktails induced more efficient exon 10 skipping than single AO treatments (Figure 3A). The lead sequences revealed by the AO screening pipeline were synthesised as PMOs (Table 1), since this chemistry performs better both in vitro and in vivo when compared to 2′-Me PS AOs [20,23,25]. 

To investigate the effect of exon 10 skipping and consequent modification of ataxin-3 protein, SCA3 patient fibroblasts were transfected with the lead candidate cocktails, prepared as 2′-Me PS AOs or PMOs, and RNA and protein were isolated 48 h following transfection. The unrelated control AO and GeneTools control PMO were included. Gel electrophoresis showed that the 2′-Me PS AO transfection consistently produced lower levels of exon 10 skipping when compared to the PMO transfections, with the exception of cocktail 22 (Figure 3A,B). The 2′-Me PS AO cocktail 22 induced an approximate 25% higher percentage of exon 10 skipping, however, this did not translate to detectable changes in the modified ataxin-3 protein (Figure 3C). Consistent with our previous experience [26], 2′-Me PS AOs have a modest ability to generate the truncated ataxin-3 isoform in vitro, with cocktail 21 resulting in about 20% of the total ataxin-3 protein being truncated (Figure 3C). Most of the 2-Me PS AO transfections downregulated ataxin-3 protein production, which was evident when normalised to the sample from the control AO transfection (Figure 3C).

In contrast, the PMOs induced downregulation of both expanded and non-expanded proteins when the data was normalised to the full-length isoforms from cells transfected with the Gene Tools control PMO (*p* value < 0.001) (Figure 3D). Downregulation of the full-length isoform was greatest after transfection with cocktail 21, with an average 5.49-fold reduction of full-length isoforms (Figure 3C,D). While full-length ataxin-3 downregulation was induced by all PMO cocktails, the truncated isoform was the predominant isoform when compared to the both full-length proteins (Figure 3C,D). Although an exact comparison may seem biased towards PMO treatments, due to the 50-fold discrepancy in transfection concentrations between the two chemistries, 2′-Me PS AO are negatively charged, and thus can be complexed with a lipid-based transfection agent (Lipofectamine 3000) for efficient cell uptake at much lower concentrations. The PMOs are uncharged and cannot be delivered into the cells under equivalent conditions. 

### 2.4. 2′-O-Methyl PS AOs Induce Sequence Independent Sequestration of Paraspeckle Protein NONO

Recently, a number of reports have described sequence-independent off-target effects associated with the phosphorothioate backbone, including non-specific interactions resulting in recruitment of paraspeckle and other nuclear proteins and formation of paraspeckle-like structures [12,15]. To establish if the 2′-Me PS ataxin-3 specific AOs can alter subcellular protein distribution, or induce nuclear inclusions, primary healthy human fibroblasts were transfected for 24 h and subsequently immunolabelled for the paraspeckle protein, NONO. Three *ATXN3* specific AOs, as well as the commercially available Gene Tools control AO, were tested as 2′-Me PS AOs and as PMOs to determine if any off-target effects are backbone specific and/or sequence dependent (Figure 4). 

Immunofluorescent staining of NONO showed that all 2′-Me PS AOs sequestrated NONO, at various subcellular locations (Figure 4A), while the identical sequences synthesised as PMOs had no such effects (Figure 4B). Consistent with the report by Flynn et al. (2018) [18], the Gene Tools control sequence seemed to induce the highest amount of NONO-containing nuclear inclusions when applied as a 2′-Me PS AO. All 2′-Me PS sequences tested in this study did induce NONO inclusions in most cells, some inclusions were located within the nucleus while others appear to be peri-nuclear or distributed in the cytoplasm (Figure 4A). This suggests that the PS backbone is directly responsible for the sequestration of NONO, while such effects were not observed when the PMO chemistry was used at higher concentrations over the same time period. 

In addition, we compared the efficiencies of exon 10 skipping mediated by the PMOs shown in Figure 4. The H10A (+ 03 + 22) AO that induced *ATXN3* exon 10 skipping was reported by Toonen et al. (2017) [8], while the other two PMOs were identified in this study (Figure 5). It can be seen that H10A (+ 10 + 34) out-performed H10A (+ 03 + 22) and H10A (+ 35 + 59) at both 10 and 5 μM concentrations (Figure 5). Although H10A (+ 03 + 22) was synthesised as a PMO, the study conducted by Toonen et al. (2017) used the MOE chemistry and thus a direct comparison cannot be made to the published data.

## 3. Discussion

The pathogenesis of SCA3 is attributed to the expanded polyQ tract that confers a toxic gain of function to the protein [4,27,28]. Therefore, the removal of the polyQ tract may provide a therapeutic strategy to delay onset or reduce severity of SCA3. As a consequence of exon 10 removal, the induced ataxin-3 isoform is missing the UIM3 domain. As reported by van Roon-Mom and colleagues, the 291aa, 34kDa protein, although missing the UIM3 domain, still binds ubiquitin in a similar manner to the full-length ataxin-3 [8]. Additionally, two naturally occurring isoforms lacking UIM3 have been shown to actually have higher rates of deubiquitination, relative to the major isoform containing the UIM3 [29]. With recent FDA approvals of AO therapeutics to restore gene expression to treat spinal muscular atrophy and DMD, pre-mRNA splicing intervention could be applied to downregulate or modify expression of toxic-gain-of-function diseases. This proof of concept study may be relevant to other polyQ diseases where the polyQ tract is found in a removable and dispensable exon. However, this would need to be the subject of functional studies to determine the effect of removing regions of other polyQ proteins. For example, Huntington’s disease is unlikely to be amenable to exon skipping, as the polyQ tract is located in the initial exon [3]. Here, we show removal of *ATXN3* exon 10, and consequently the polyQ repeat from the normal and expanded ataxin-3 protein, using two different AO chemistries. Additionally, these in vitro experiments demonstrate proof of concept and can provide pre-clinical candidate molecules for eventual in vivo administration. Dosage regimens, clearance rates and delivery methods will need to be optimised to assess the effects of PMOs compared to other chemistries.

We confirm apparent off-target effects of AOs on a PS backbone, transfected in vitro. The 2′-Me PS AO targeting *ATXN3* induced sequestration of the paraspeckle protein NONO, while no such consequences were observed when cells were treated with the same sequences synthesised as PMOs, in keeping with our previously reported findings and those of the Crooke group [12,13,18]. While these studies were only conducted in vitro, van Roon-Mom and colleagues described activation of the innate immune system as the result of intracerebroventricular administration of 2′-Me AOs into mice [15]. Marked upregulation of Oasl2 and Bst2 proteins, both of which are involved in interferon signalling, particularly in response to viral infections [30,31,32], was a major finding [15]. Several other studies have described adverse effects of AOs on a PS backbone, including thrombocytopenia, severe injection site reaction, cytotoxicity and induction of double-stranded DNA breaks [33,34,35]. 

The safety of PMOs in the clinic is evident from long term treatment (240 weeks) of DMD patients with the PMO *Eteplirsen*. No evidence of serious adverse reactions were reported [33,36], while the same cannot be said for the 2′-Me PS AO drug *Drisapersen.* Every drug on a the PS backbone tested to date has been reported to elicit off-target effects [33,36,37]. This may be explained by the crucial difference that PMOs, unlike 2′-Me PS AOs, are uncharged and therefore do not readily interact with proteins [38,39,40]. We, and others have shown that PMOs generally outperform 2′-Me PS AOs for splice switching applications [37,41], possibly due to the superior stability, specificity and binding affinity of PMOs, compared to 2′-Me PS AOs [38,40]. Although PMO-treated cells were transfected at higher concentrations than the 2′-Me PS AOs that are transfected as lipoplexes, these high concentrations are required for gymnotic PMO uptake but were nevertheless well tolerated by cells. Although the delivery of 2′-Me PS AOs is greatly enhanced through the use of cationic liposome preparations, concentrations of 600 nM and above lead to widespread cell death in vitro.

Although PMOs outperformed 2′-Me PS AOs in vitro, the PMOs do have several limitations. When administered in vivo, PMOs are rapidly cleared by the renal system and must therefore be dosed at relatively high levels. This is compounded by poor uptake and delivery of PMOs into target tissues [42]; due to their uncharged nature PMOs cannot readily move through cellular membranes [43]. These limitations are currently subject to extensive research in developing technologies, such as conjugation to cell penetrating peptides and various physical and chemical methods to increase uptake and reduce the rapid clearance [42,44]. Despite these draw backs, from our in vitro data we believe PMOs to have greater therapeutic potential for SCA3 and other diseases, such as muscular disorders that are amenable to splice switching [45]. In saying that, it is important to note that in vitro studies and in vivo animal models do not always translate into successful treatments for patients. This is mainly due to sequence differences in the cases of in vivo models, with a prime example being a mouse model of Duchenne muscular dystrophy, whereby subtle changes in sequences can drastically affect AO efficiency [46]. 

We show significant knockdown of the full-length ataxin-3 isoforms encoded by both alleles, as well as modification of the ataxin-3 protein to produce a truncated protein, missing the polyQ tract. While the impact of global ataxin-3 knockdown is under debate [8,47,48], the role of ataxin-3 in the ubiquitin-proteasome machinery is well-established, and there are conflicting views as to whether ataxin-3 is essential in maintaining normal cellular function [3,49,50]. Figiel and colleagues (2011) created a functional *Atxn3* knockout mouse that showed no obvious phenotype, with a life span comparable to that of the wildtype mouse [48]. Another group reported similar findings, with no adverse effects on *Atxn3* knockout mouse life span or fertility and no apparent abnormalities, but they did report apparent increased anxiety and increased levels of ubiquitinated proteins in the *Atxn3* knockout model [51]. Separately, AO treated transgenic SCA3 mice (expressing a human full-length, 84Q *ATXN3* gene), showed that knockdown of ataxin-3 was well tolerated with no signs of astrogliosis or microgliosis [52]. However, our data show a high proportion of the truncated protein as a consequence of *ATXN3* exon 10 skipping, and thus this may provide sufficient functional protein to support ubiquitination and protein degradation [53]. With that being said, the Paulson group conducted in vitro experiments to assess the effects of ATXN3 knockout using *Atxn3* null mouse embryonic stem cells. They found that the loss of Atxn3 caused dysregulation in signalling pathways that included depression of Wnt and BMP4 pathways, as well as elevated growth factor pathways [54]. In contrast, the same group showed that knockdown of an expanded *ATXN3* in a transgenic mouse model (MJD-Q84.2) using a 2′-MOE AO rescued the phenotype with no apparent adverse effects, thus suggesting in vivo treatment and knockdown of ATXN3 may be feasible [55]. Moreover, in the current study it is believed that the removal of the CAG repeat alone without removal of key functional domains would result in limited downstream effects. Further studies investigating the long-term impact of ataxin-3 knockdown in vivo will be required to determine if skipping of exon 10 has potential as a treatment for SCA3.

In conclusion, while this study was a preliminary in vitro investigation, PMOs consistently produced a significantly higher proportion of the truncated protein, missing the toxic polyQ repeat, relative to 2′-Me PS AOs. With increasing numbers of AO therapeutics being approved for clinical use, our results suggest that the lead PMOs may be an attractive therapeutic option for the treatment of spinocerebellar ataxia type 3.

## 4. Materials and Methods 

### 4.1. AO Design and Synthesis

Splice-switching AOs were designed to target and anneal to splicing motifs at the intron/exon boundaries as well as predicted exon splice enhancer sequences identified using the web-based application, *Human Splicing Finder 3.0* [56]. In addition, specificity of the AOs for the *ATXN3* target motifs was confirmed via BLAST analysis to identify potential off-target annealing. 2′-Me PS AOs were obtained from TriLink Biotechnologies (Maravai LifeSciences, San Diego, CA, USA), while PMOs were purchased from Gene Tools, LLC (Philomath, OR, USA). Nomenclature of AOs is according to Aung-Htut, McIntosh et al. (2019) and indicates gene, exonic target with annealing coordinates relative to the intron:exon:intron arrangement (Table 1 and Table 2) [57].

### 4.2. Cell Culture

Primary dermal fibroblasts were cultured from a skin biopsy taken from a healthy volunteer, after informed consent, and the project received approval from the Human Research Ethics Committee at Murdoch University (approval number, 2013/156). Healthy human fibroblasts were cultured in Dulbecco’s Modified Essential Medium (Gibco; Life Technologies, Melbourne, Australia), supplemented with 15% fetal bovine serum (FBS) (Scientifix, Cheltenham, Australia). The SCA3 fibroblast cell line (GM6151b) was obtained from Coriell Cell Repositories (Camden, NJ, USA) and cultured in minimal essential medium (MEM), supplemented with 15% FBS (Scientifix), 1% Glutamax (Gibco) and 1x penicillin/streptomycin. 

### 4.3. Transfection

All cell strains were either transfected with 2′-Me PS AO, Lipofectamine 3000 (Life Technologies) lipoplexes in Opti-MEM (Gibco) according to manufacturer’s instructions, or with PMOs, using Endo-Porter (Gene Tools, LLC) according to manufacturer’s instructions in MEM (Gibco), supplemented with 7.5% FBS (Scientifix) [58]. Cells were harvested 24 h following transfection for transcript analysis or after 48 h for protein studies. 

### 4.4. RNA Extraction and RT-PCR Assays

Total RNA was extracted using the MagMax™ nucleic acid isolation kits (ThermoFisher Scientific, Melbourne Australia) in accordance with the manufacturer’s instructions. Transcripts were amplified using the one-step SuperScript^®^ III reverse transcriptase, with 50 ng of total RNA as the template. To amplify the *ATXN3* transcript, exon 7-F (5′GTCCAACAGATGCATCGACCAA3′) and exon 11-R (5′AGCTGCCTGAAGCATGTCTTCTT3′) primers were used (Gene Works, Adelaide, Australia). The cycling reactions included 55 °C for 30 min, 94 °C for 2 min, with 28 cycles of 94 °C 30s, 55 °C 30s and 68 °C 1.5 min. The PCR products were fractionated on 2% agarose gels in Tris-Acetate-EDTA buffer.

### 4.5. Western Blotting

Cell lysates (~800,000 cells) were prepared in 100 μL of 125 mM Tris-HCl, pH 6.8, 15% SDS, 10% Glycerol, 1.25 µM PMSF (Sigma-Aldrich, Sydney, Australia) and 1 × protease inhibitor cocktail (Sigma-Aldrich) and subsequently sonicated 6 times (1 s pulses) prior to the addition of bromophenol blue (0.004%) and dithiothreitol (2.5 mM). Samples were heated at 94 °C for 5 min, cooled on ice and centrifuged at 14,000× *g* for 2 min before loading onto the gel.

Total protein (25 µg), determined by a BCA assay (ThermoFisher Scientific), was loaded onto NuPAGE Novex 4–12% Bis/Tris gradient gels (ThermoFisher Scientific). Samples were subsequently fractioned at 200 volts for one hour. Proteins were then transferred onto a Pall Fluoro Trans^®^ polyvinylidene fluoride membrane at 350 mA for one hour. Following blocking in 5% skim milk in TSBT for one hour, the membrane was incubated with either mouse monoclonal anti-ataxin-3 antibody (Millipore, cat. No. MAB5360, Billerica, USA) at 1:500 dilution or rabbit polyclonal anti-β-tubulin antibody (ThermoFisher Scientific, cat. No. PA1-41331) at 1:1000 dilution in 5% skim milk in TSBT, overnight at 4 °C. For detection of pathogenic polyQ stretches, the membrane was probed with mouse monoclonal anti-polyQ antibody (Millipore, cat. no. MAB1574, Billerica, USA), at 1:1000 dilution in 5% skim milk in TSBT, overnight at 4 °C. 

For immunodetection, polyclonal goat anti-rabbit or anti-mouse immunoglobulins/HRP (Dako, cat. no P0448 and D0447 respectively, Sydney, Australia) at a dilution of 1:10,000 and Luminata Crescendo Western HRP substrate (Merk Millipore, Sydney, Australia) were used. The blots were exposed for a serial scan of 20 s using the Fusion FX gel documentation system (Vilber Lourmat, Marne-la-Vallée, France).

### 4.6. Immunofluorescence 

Approximately 7500 patient fibroblasts were seeded into each well of an 8 well chamber slide (Ibidi, Martinsried, Germany) and incubated for 24 h, prior to transfection. Following transfection, with 2′-Me PS AOs or PMOs, cells were fixed in ice-cold acetone:methanol (1:1) for 5 min and then air dried.

Fixed cells were incubated in PBS containing 1% Triton X-100 for 10 min at room temperature to permeabilise the nuclear membrane, and then in PBS to remove excess Triton X-100. Mouse anti-NONO monoclonal antibody (a gift from Prof. Archa Fox, The University of Western Australia) was diluted in PBS containing 0.05% Tween20 and applied to cells for one hour at room temperature. NONO was detected using AlexaFluor488 anti-mouse IgG (ThermoFisher Scientific, cat no. A-11001) (1:400) after incubation for one hour at room temperature, and subsequently counterstained with Hoechst 33342 (Sigma-Aldrich) for nuclei detection (1 mg/mL diluted, 1:125).

### 4.7. Densitometric and Statistical Analysis

Densitometric analysis was conducted using ImageJ (version 1.8.0_112) imaging software (NIH, Bethesda, MD, USA) [59]. *p* value refers to unpaired two tailed student’s t tests. A *p* value < 0.05 was considered statistically significant. 

## Figures and Tables

**Figure 1 ijms-20-05434-f001:**
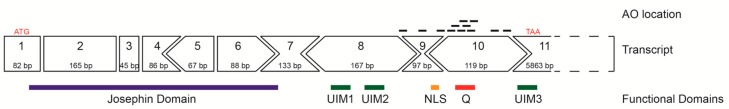
Schematic representation of the *ATXN3* gene transcript (ENST00000558190.6) and reading frame, showing location of encoded protein (361 amino acid) domains below the exon map. In-frame exons are represented as rectangles, whereas those bounded by partial codons are represented with chevron sides. Exonic/intronic locations of antisense oligonucleotides designed to redirect *ATXN3* pre-mRNA processing are represented as black bars above the transcript.

**Figure 2 ijms-20-05434-f002:**
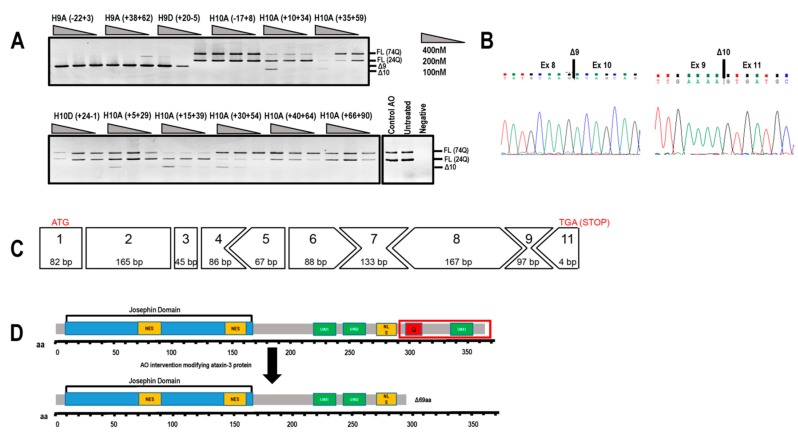
Evaluation of antisense oligonucleotides (AOs) designed to alter *ATXN3* transcript structure. (**A**) Screening of 2′-Me phosphorothioate (PS) AOs targeting exon 9 or exon 10 for removal. AOs were transfected as lipoplexes at three concentrations, 400, 200, 100 nM, and after RT-PCR and gel fractionation, products representing the full-length (FL 74Q; FL 24Q), exon 9-skipped (Δ9, 384 bp) and exon 10-skipped (Δ10, 362bp) transcripts were identified. (**B**) Sanger sequencing of the RT-PCR products generated by skipping of exon 9 or 10, shows the junction between exon 8 and 10, and exons 9 and 11, respectively. (**C**) Although removal of exon 10 alters the reading frame, the residue encoded by the exon 9 and 11 junction codon is synonymous (lysine). Immediately following this lysine is an in-frame termination codon (TGA). (**D**) Removal of exon 10 from the *ATXN3* coding sequence results in a truncated, 291 aa protein (missing the terminal 69 aa) of approximately 34 kDa that is reported to be functional. aa = amino acid; NES = nuclear export signal; NLS = nuclear localisation signal; UIM = ubiquitin interacting motif. Q = polyglutamine tract.

**Figure 3 ijms-20-05434-f003:**
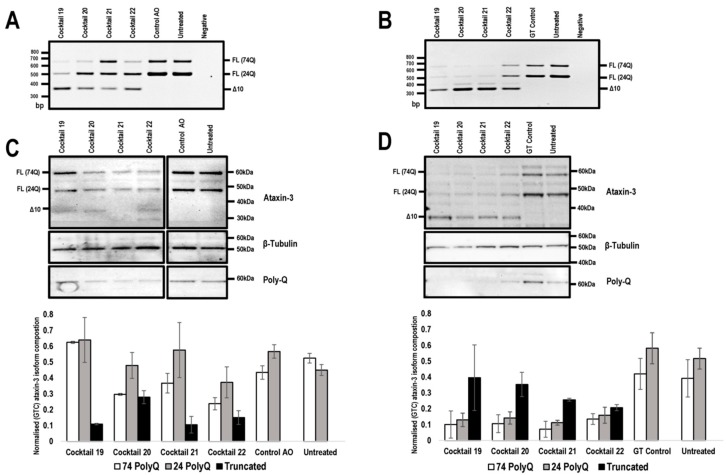
Comparison of *ATXN3* exon 10 skipping in patient cells, transfected with 2′-Me PS AOs and PMOs. Cells were harvested 48 hr following transfection for protein and RNA analysis. Agarose gel fractionation of *ATXN3* amplicons, following 2′-Me PS AO (**A**) and PMO (**B**) cocktail transfections at concentrations of 400 nM and 20 μM, respectively shows full-length (FL 74Q; FL 24Q) and induced transcript products after skipping of exon 10 (Δ10). Ataxin-3 protein was analysed by Western blotting following 2′-Me PS AO (**C**) and PMO (**D**) cocktail transfection at a concentration of 400 nM and 20 μM, respectively. The disease-causing 74Q protein is approximately 60 kDa, the protein encoded by the healthy allele is approximately 48 kDa and the Δ10 encoded protein, 34 kDa. Beta-Tubulin was used as a loading control. The samples were also probed with an anti-polyglutamine antibody to identify the pathogenic stretch of glutamines in the ataxin-3 protein. Densitometric analysis performed on the Western blots are shown below the blots (means plus error bars. Error bars = standard deviation, *n* = 3). Samples are normalised to the GT control. (FL = full-length, Δ10 = exon skipped product, GT = Gene Tools control PMO, Q = glutamine, PolyQ = polyglutamine).

**Figure 4 ijms-20-05434-f004:**
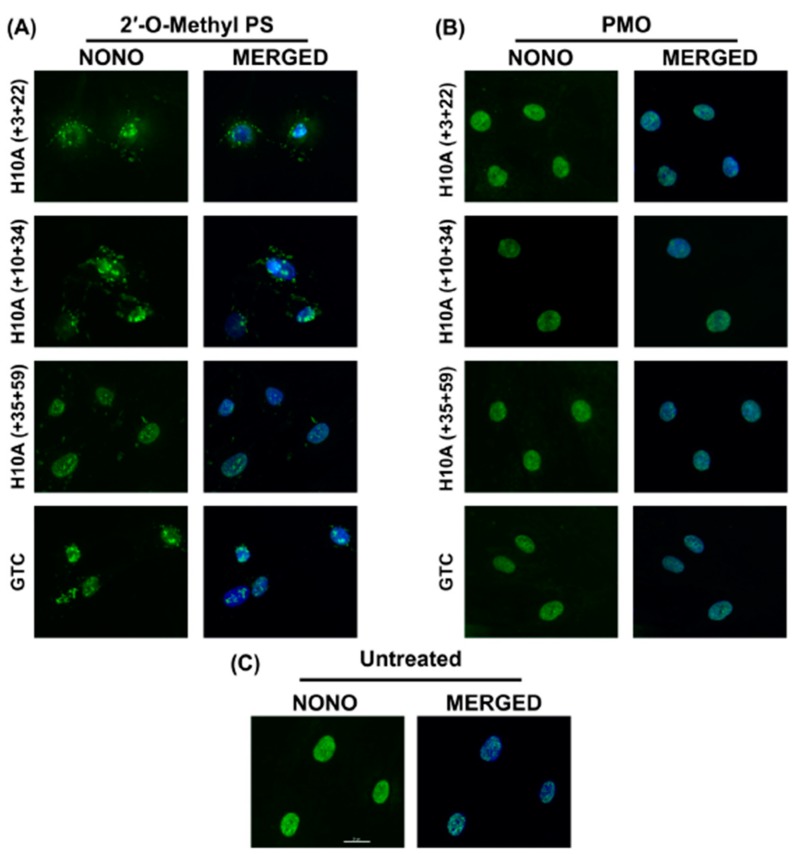
Immunofluorescent labelling of paraspeckle protein NONO in transfected healthy fibroblasts. (**A**) Immunolabelling of primary normal human fibroblasts following transfection with various 2′-Me PS AO lipoplexes at a concentration of 400 nM. Cells were fixed 24 hrs following transfection. Staining of the paraspeckle protein NONO shows sequestration and aggregation of NONO in various subcellular locations. (**B**) Immunofluorescent labelling of NONO following Endoporter transfection of primary normal human fibroblasts with various PMOs at concentrations of 10 or 5 μM. Cells were fixed 24 hrs following transfection. (**C**) Untreated (control) showing endogenous subcellular distribution of NONO. GTC = Gene Tools Control sequence. Scale bar is 20 μm.

**Figure 5 ijms-20-05434-f005:**
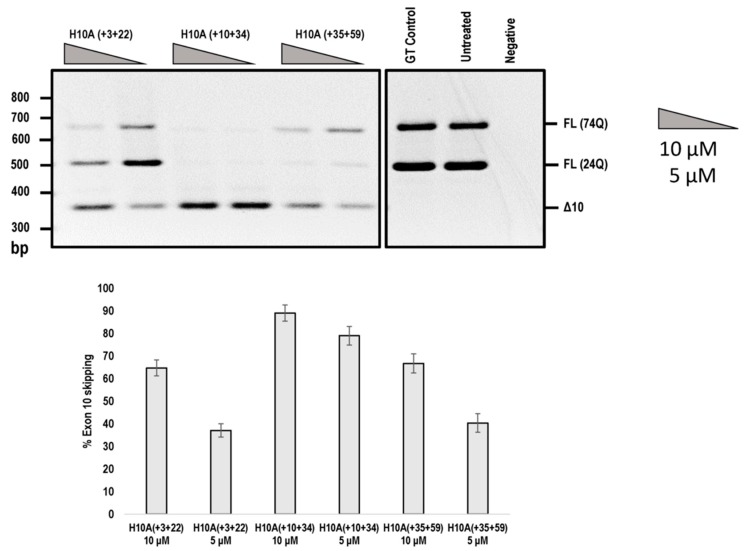
Agarose gel fractionation of *ATXN3* transcript products from SCA3 patient derived fibroblasts, following PMO transfection at concentrations of 10 and 5 μM. Fractionation shows full-length products (FL 74Q; FL 24Q) and removal of exon 10 (Δ10) (product size of 362 bp). Densitometric analysis of the gel image is shown below the gel (means plus error bars. Error bars = standard deviation, *n* = 3). (FL= full-length, Δ10 = exon skipped product, GT = Gene Tools control PMO, Q = glutamine).

**Table 1 ijms-20-05434-t001:** Sequences and coordinates of AOs employed in this study.

Name	Sequence (5′–3′)
ATXN3 H9A (− 22 + 03)	UACCUGAAAACAAAACACAACACAA
ATXN3 H9A (+ 38 + 62)	UUCUGAAGUAAGAUUUGUACCUGAU
ATXN3 H9D (+ 20 − 05)	UUUACUUUUCAAAGUAGGCUUCUCG
ATXN3 H10A (− 17 + 08)	GCUGCUGUCUGAAACAUUCAAAAGU
ATXN3 H10A (+ 10 + 34) *	CUGCUGCUGCUGCUGUUGCUGCUUU
ATXN3 H10A (+ 35 + 59) *	GUCCUGAUAGGUCCCCCUGCUGCUG
ATXN3 H10D (+ 24 − 01) *	CCUAGAUCACUCCCAAGUGCUCCUG
ATXN3 H10A (+ 05 + 29)	GCUGCUGCUGUUGCUGCUUUUGCUG
ATXN3 H10A (+ 15 + 39) *	UGCUGCUGCUGCUGCUGCUGUUGCU
ATXN3 H10A (+ 30 + 54)	GAUAGGUCCCCCUGCUGCUGCUGCU
ATXN3 H10A (+ 40 + 64)	ACUCUGUCCUGAUAGGUCCCCCUGC
ATXN3 H10A (+ 66 + 90)	GUGGCUGGCCUUUCACAUGGAUGUG
ATXN3 H10A (+ 03 + 22) #^/^*	CUGUUGCUGCUUUUGCUGCU
Control AO ^@^	GGAUGUCCUGAGUCUAGACCCUCCG
Gene Tools Control	CCTCTTACCTCAGTTACAATTTATA

* These sequences were selected for synthesis as the phosphorodiamidate morpholino oligomer chemistry (PMO). PMO oligomers are synthesised with Thymine (T) rather than Uracil (U). # Sequence from Toonen et al. (2017) [8]. ^@^ unrelated sham control sequence. GeneTools Control; commercially available from GeneTools.

**Table 2 ijms-20-05434-t002:** AO cocktails containing two different, non-overlapping exon 10 sequences. Each cocktail comprises of a 1:1 molar ration of each AO.

Cocktail Number	AO Combination
Cocktail 19	ATXN3 H10A (+ 10 + 34)
	ATXN3 H10A (+ 35 + 59)
Cocktail 20	ATXN3 H10A (+ 10 + 34)
	ATXN3 H10D (+ 24 − 01)
Cocktail 21	ATXN3 H10A (+ 35 + 59)
	ATXN3 H10D (+ 24 − 01)
Cocktail 22	ATXN3 H10A (+ 15 + 39)
	ATXN3 H10D (+ 24 − 01)
Cocktail 23	ATXN3 H10A (+ 05 + 29)
	ATXN3 H10D (+ 24 − 01)
Cocktail 24	ATXN3 H10A (+ 30 + 54)
	ATXN3 H10D (+ 24 − 01)

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
