# Peer review of "Removal of the Polyglutamine Repeat of Ataxin-3 by Redirecting pre-mRNA Processing"

_ijms, 2019, doi:10.3390/ijms20215434_

Round 1

Reviewer 1 Report

The manuscript presented by McIntosh and coworkers demosntrate that the efficiency in the removal of the toxic polyglutamine tract of ataxin-3 in vitro is improved using phosphorodiamidate morpholino oligomers. These atoxic morpholino oligomers may provide greater therapeutic benefit for the treatment of spinocerebellar ataxia type 3, when comparing to to antisense oligonucleotides composed of 2′-O-methyl modified bases on a phosphorothioate backbone.

The manuscript provides novel and relevant insights of a potential therapeutic treatment for spinocerebellar ataxia type 3. I have only one comment to suggest in order to improve the manuscript. The authors clearly discuss the advantages and limitations of using phosphorodiamidate morpholino oligomers in vivo. In this sense, a more detailed explanation of this point should be interesting to include in the Discussion section. The use of this oligomers can open the door to more eficient therapeutic strategies in different diseases, as an example in muscular disorders (Aoki Y et al., doi: 10.1093/hmg/ddt341). This is interesting to discuss since the majority of the treatments tested in vitro and in vivo in animal models can't be successfully identified as a treatment in patients.

Author Response

Point 1: I have only one comment to suggest in order to improve the manuscript. The authors clearly discuss the advantages and limitations of using phosphorodiamidate morpholino oligomers in vivo. In this sense, a more detailed explanation of this point should be interesting to include in the Discussion section.

The use of this oligomers can open the door to more efficient therapeutic strategies in different diseases, as an example in muscular disorders (Aoki Y et al., doi: 10.1093/hmg/ddt341). This is interesting to discuss since the majority of the treatments tested in vitro and in vivo in animal models can't be successfully identified as a treatment in patients.

Response 1:  We have included in the discussion additional references describing application of PMOs to other conditions (Aoki Y et al., doi: 10.1093/hmg/ddt341) – reference 45and Miyatake et al. (2018) – reference 46.

Lines 332 – 338

“Despite these draw backs, from our in vitro data we believe PMOs to have greater therapeutic potential for SCA3 and other diseases, such as muscular disorders that are amenable by splice switching AOs [45]. With saying that, it is important to note that in vitro studies and in vivo animal models do not always transfer as successful treatments to patients. This is mainly due to sequence differences in the cases of in vivo models, with a prime example being the mdx mouse model of Duchenne muscular dystrophy. Whereby subtle changes in sequences can drastically affect the efficiencies of AOs [46].” 

Reviewer 2 Report

Spinocerebellar ataxia type 3 (SCA3) is a condition characterized by progressive problems with movement. This condition is inherited in an autosomal dominant pattern with one copy of the altered gene in each cell sufficient to cause the disorder. Short cytosine-adenine-guanine repeat expansions (CAG) are implicated in seven other neurodegenerative disorders and all disorders are caused by CAG repeat expansion and translated into a stretch of polyglutamines in the respective proteins. The authors report the improved efficiency in the removal of the toxic polyglutamine tract of ataxin-3 in vitro using phosphorodiamidate morpholino oligomers. Downregulation of both the expanded and non-expanded protein was induced by the morpholino antisense oligomer, with a greater proportion of ataxin-3 protein missing the polyglutamine tract. The authors indicate that the use of a phosphorodiamidate morpholino oligomer may be safer for clinical application than other antisense oligonucleotides therapy and provide a better treatment of spinocerebellar ataxia type 3 for co-ordination and balance problems and other early signs and symptoms.

Minor Comment:

An abbreviation list may be required.

Questions:

Q1. Is the use of the improved efficiency of removal of toxic polyglutamine tract of ataxin 3 relevant to other neurodegenerative diseases with relevance to CAG repeat/polyglutamine tract diseases?

Q2. Can the authors compare their in vitro data and conduct in vivo studies (spinocerebellar ataxia type 3 mice) with their phosphorodiamidate morpholino oligomers and compare their results to an established efficacy of ATXN3-targeted antisense oligonucleotides as a disease-modifying therapeutic strategy for SCA3?

Q3. Can the removal of the toxic polyglutamine tract of ataxin-3 using phosphorodiamidate morpholino oligomers be associated with alterations in other ATXN3 associated multiple signal transduction pathways (protein aggregation, ubiquitin-proteasomal degradation, transcriptional dysregulation)?

Q4. Are the use of phosphorodiamidate morpholino oligomers in these experiments better than antisense oligonucleotides with relevance to doses used, clearance rates and cell nuclear/ cytoplasmic accumulation and binding to lipids versus proteins in cells?

RELEVANT REFERENCES:

Koshy BT, Zoghbi HY. The CAG/polyglutamine tract diseases: gene products and molecular pathogenesis. Brain Pathol. 1997 Jul;7(3):927-42. Rudnicki DD, Margolis RL. Repeat expansion and autosomal dominant neurodegenerative disorders: consensus and controversy. Expert Rev Mol Med. 2003 Aug 22;5(21):1-24. McLoughlin HS, et al. Oligonucleotide therapy mitigates disease in spinocerebellar ataxia type 3 mice. Ann Neurol. 2018 Jul;84(1):64-77. Zeng L, Zhang D, McLoughlin HS, Zalon AJ, Aravind L, Paulson HL. Loss of the Spinocerebellar Ataxia type 3 disease protein ATXN3 alters transcription of multiple signal transduction pathways. PLoS One. 2018 Sep 19;13(9):e020443

Author Response

Minor Comment – An abbreviation list may be required.

Response – Agreed and an abbreviation list has been added

Point 1:Is the use of the improved efficiency of removal of toxic polyglutamine tract of ataxin 3 relevant to other neurodegenerative diseases with relevance to CAG repeat/polyglutamine tract diseases?

Response 1:Yes, we believe so as the removal of CAG repeat region is feasible for other disease where the CAG repeat is located in an exon that can be skipped.

This is elaborated in the discussion Lines 291 – 296. “This proof of concept study may be relevant to other polyQ diseases where the polyQ tract is found in a removable and dispensable exon. However, this would need to be the subject of functional studies to determine the effect of removing regions of other polyQ proteins. For example, Huntington’s disease would potentially not be amenable via this traditional exon skipping, as the polyQ tract is located in the initial exon [3].

Point 2:Can the authors compare their in vitro data and conduct in vivo studies (spinocerebellar ataxia type 3 mice) with their phosphorodiamidate morpholino oligomers and compare their results to an established efficacy of ATXN3-targeted antisense oligonucleotides as a disease-modifying therapeutic strategy for SCA3?

Response 2:The reviewer raises an excellent point for future studies but unfortunately due to funding and time constraints these studies are not feasible at this stage. In addition, the intellectual property landscape is such that we do not have access to the 2’-MOE chemistry and therefore cannot conduct direct chemistry comparison experiments. It was for this reason that the 2’-O-Me chemistry was used in this study rather than the previously reported 2’-MOE oligos.

Point 3:Can the removal of the toxic polyglutamine tract of ataxin-3 using phosphorodiamidate morpholino oligomers be associated with alterations in other ATXN3 associated multiple signal transduction pathways (protein aggregation, ubiquitin-proteasomal degradation, transcriptional dysregulation)?

Response 3:As the truncated protein has been shown to be functional (as cited in the manuscript), we believe there would not be a significant effect on downstream pathways. This would be attributed to the CAG repeat region not being associated with any functional pathways of ATXN3.

We have elaborated on this topic in the discussion of the manuscript Lines 352 – 360  

“With that being said, the Paulson group conducted in vitroexperiments to assess the effects of ATXN3 knockout using Atxn3 null mouse embryonic stem cells. They found that the loss of Atxn3 caused dysregulation in signalling pathways that included depression of Wnt and BMP4 pathways, as well as elevated growth factor pathways [52]. In contrast the same group showed that knockdown of an expanded ATXN3 in a transgenic mouse model using a 2′-MOE AO rescued the phenotype with no apparent adverse effects, thus suggesting in vivo treatment and knockdown of ATXN3 may be feasible [53]. Moreover, in the current study it is believed that the removal of the CAG repeat alone without removal of key functional domains would result in limited downstream effects.”

Point 4:Are the use of phosphorodiamidate morpholino oligomers in these experiments better than antisense oligonucleotides with relevance to doses used, clearance rates and cell nuclear/ cytoplasmic accumulation and binding to lipids versus proteins in cells?

Response 4:

We are unsure of the questions being raised here. We have mentioned in the discussion factors related to accumulation regarding paraspeckle proteins (Lines 301 – 311)

We had added into the discussion Lines 297 – 300  

“Additionally, these in vitro experiments demonstrate proof of concept and can provide pre-clinical candidate molecules for eventual in vivo administration. Regimen doses, clearance rates and delivery methods will need to be optimised to assess the effects of PMOs compared to other chemistries.”

Point 5:Relevant references.

Response 5:These references were added and discussed in the discussion with regards to Points 1 – 4.

Round 2

Reviewer 2 Report

The authors have made modifications to the revised manuscript.

The revised manuscript maintains the high standards for peer-reveiwed journals and is appropriate for scholarly publication.